# Comparison of Experienced and Novice Drivers’ Visual and Driving Behaviors during Warned or Unwarned Near–Forward Collisions

**DOI:** 10.3390/s23198150

**Published:** 2023-09-28

**Authors:** Jordan Navarro, Emanuelle Reynaud, Marie Claude Ouimet, Damien Schnebelen

**Affiliations:** 1Laboratoire d’Etude des Mécanismes Cognitifs (EA 3082), University Lyon 2, 69007 Lyon, France; emanuelle.reynaud@univ-lyon2.fr (E.R.); damien.schnebelen@univ-lyon2.fr (D.S.); 2Institut Universitaire de France, 75005 Paris, France; 3Faculté de Médecine Et des Sciences de La Santé, Université de Sherbrooke, Sherbrooke, QC J1H 5N4, Canada; marie.claude.ouimet@usherbrooke.ca

**Keywords:** human–machine cooperation, warning, forward collision warning system, visual behaviors, steering behaviors, car driving, car-following task

## Abstract

Forward collision warning systems (FCWSs) monitor the road ahead and warn drivers when the time to collision reaches a certain threshold. Using a driving simulator, this study compared the effects of FCWSs between novice drivers (unlicensed drivers) and experienced drivers (holding a driving license for at least four years) on near-collision events, as well as visual and driving behaviors. The experimental drives lasted about six hours spread over six consecutive weeks. Visual behaviors (e.g., mean number of fixations) and driving behaviors (e.g., braking reaction times) were collected during unprovoked near-collision events occurring during a car-following task, with (FCWS group) or without FCWS (No Automation group). FCWS presence reduced the number of near-collision events drastically and enhanced visual behaviors during those events. Unexpectedly, brake reaction times were observed to be significantly longer with FCWS, suggesting a cognitive cost associated with the warning process. Still, the FCWS showed a slight safety benefit for novice drivers attributed to the assistance provided for the situation analysis. Outside the warning events, FCWS presence also impacted car-following behaviors. Drivers took an extra safety margin, possibly to prevent incidental triggering of warnings. The data enlighten the nature of the cognitive processes associated with FCWSs. Altogether, the findings support the general efficiency of FCWSs observed through a massive reduction in the number of near-collision events and point toward the need for further investigations.

## 1. Introduction

When considering human cognitive abilities, researchers tend to focus their efforts on the humans themselves. However, in practice, a vast number of our daily cognitive abilities are mediated using an ever-larger body of technologies. Consequently, human–technology relationships comprise a major topic of interest for both the understanding of technology use and human cognitive processing. In theory, it has been proposed that human–machine relationships are symbiotic in such a way that humans both live and evolve in a world where technology shapes their cognition [1]. In practice, this theoretical framework can be applied to a large set of everyday situations, such as writing using a text editor, planning professional activities with an agenda, and monitoring time with a clock.

Among all human activities, mobility is a pillar of our species, from its origin with bipedalism to the most technologically advanced mobility solutions. Car driving is a well-spread daily activity in modern societies and, with regular technological advances, constitutes a highly relevant investigation framework for human–machine relationships [2,3,4]. Although driving is an activity experienced as unique, it is composed of a large set of sub-tasks [5,6] that require a variety of cognitive processes. These sub-tasks have been regrouped into three main classes [7], engaging distinct neural networks [8] and allowing drivers to (i) operate the vehicle based on (ii) tactical decisions made taking into account the current driving context and depending on (iii) strategical driving plans.

Among the different technologies developed, audio and visual driver warning systems have become a classic in-vehicle technology. For example, a driver can be warned because their seatbelt is not fastened, the headlights of their vehicle are not turned on, or their door is not closed properly. Such warnings can be described as reminders to supplement drivers’ memory capabilities. Other warnings used are designed to help drivers maintain optimal steering performances by assisting the visual and attentional processes engaged while driving. Most of the automation delivering warnings target the driving tasks referred to as operational control, i.e., the driving abilities dedicated to vehicle guidance. Operational control can be subdivided into longitudinal (i.e., speed adjustments) and lateral control (i.e., steering adjustments). Similarly, warning devices have been designed to prevent forward collisions (FCWSs—forward collision warning systems) and lane departures (LDWSs—lane departure warning systems). FCWSs monitor the road ahead and warn drivers when the time to collision with the leading vehicle reaches a certain threshold. LDWSs also monitor the road ahead and warn drivers when the vehicle reaches the lane boundaries.

A model of the cognitive processes associated with LDWSs has been proposed in light of the most dominant human–machine theories in a review of the literature on empirical results [9]. Here, the focus was set on longitudinal control and associated FCWS, a feature described as a standard to implement in vehicles based on potential cost–benefit estimations [10]. Although FCWSs rely on a similar technological basis as LDWSs (i.e., detection of an imminent hazard), they may imply different cognitive processes for the driver. Indeed, the expected behavior following an FCW is to decelerate, either by releasing the gas pedal or pressing the brake pedal. In contrast, with the LDWS, the driver is warned of an imminent lane departure, but the action to perform can be to steer the wheel either to the left or right. LDWSs urge drivers to correct their lateral position, whereas the FCWSs urge drivers to decelerate. In more experimental terms, an FCWS requires only one response with different possible magnitudes (i.e., the intensity of the deceleration), whereas an LDWS requires first a decision on which motor response to apply (i.e., turning the steering wheel to the left or to the right), followed by the selection of the magnitude (i.e., steering wheel amplitude).

The benefit of FCWSs has been repeatedly reported in the literature [11,12,13,14,15], in most cases, to prevent or attenuate collisions with a leading vehicle but also in other situations, such as car-to-cyclist collisions [16]. To detail those safety benefits, several explanations have been proposed.

The most straightforward explanation is a reduction in the time required to press the brake pedal (often referred to as the brake reaction time). The available literature, however, showed mixed results. Some studies reported shorter brake reaction times [15,17,18,19] or faster accelerator pedal releases [14,20] when FCWs were delivered, supporting the idea that the perception, process, and response to a hazardous event could improve with it. However, shorter brake reaction times were not systematically observed in other studies. In one study, FCWs were only recorded to reduce brake reaction times for large speed differences between leading and following vehicles and when the lead vehicle was accelerating [21]. In other studies, FCW only reduced brake reaction times under specific experimental conditions but not in other conditions [22], or simply, no significant brake reaction time reduction with an FCW was recorded [14]. These mixed results suggest that not all the benefits of FCWSs are to be found in faster brake reaction times.

Another explanation may be that FCWSs allow for a more adequate response to the near-collision event. In line with this idea, an FCW was observed to trigger a larger brake pedal force than unwarned events [18]. In the absence of the FCW, drivers were also observed to come closer to a collision with the lead vehicle [22]. Such warning benefits could be related to gaze redirection toward relevant elements of the driving scene [23,24,25,26].

FCWSs can also trigger behavioral adaptations outside the near-collision events warned. As FCWSs are designed to prevent forward collisions, car-following tasks have often been used to conduct experiments to test their effects. The presence of FCWSs in regular driving situations was associated with changes in car-following behaviors, either by following the leading vehicle more closely [21] or by keeping a larger distance from it [27]. Again, studies show mixed results regarding behavioral adaptations.

The discrepancies in the results in the literature can be explained by different experimental conditions, specific characteristics of the FCWS [28], or driver differences. Regarding the latter, FCWSs were observed to be effective for both aggressive and non-aggressive drivers, but only aggressive drivers judged warnings adapted to their driving style as more acceptable [22]. FCWs were also reported to speed up drowsy drivers’ responses, but only when drivers looked away from the forward roadway [29]. Driving experience also seems to play a role when facing urgent leading-vehicle braking, with drivers with less than three years of driving experience showing longer brake reaction times than drivers with more than five years of experience [30].

The current study aimed to investigate the effect of the presence of an FCWS on near-crash events and visual and driving behaviors compared to a control condition without an FCWS. Based on previous observations, FCWSs were hypothesized to reduce the number of near-collisions and improve drivers’ responses to occurring near–forward collisions. According to the LDWS model [9], it was hypothesized that with an FCW, drivers would (i) interrupt the current task, (ii) activate a motor response, and (iii) redirect their visual attention toward relevant information to deal with the near–forward collision event, resulting in faster and more efficient maneuvers. In practical terms, it was hypothesized that, compared to an unwarned near-collision, an FCW would trigger gaze redirection toward relevant information, reduce the brake reaction time, trigger sharper deceleration, and ultimately increase the safety margin with the lead vehicle (i.e., greater time to collision). FCWS benefits were anticipated to be more substantial for novices than for experienced drivers. Finally, FCWS presence was also expected to alter the quality of the car-following task in general.

## 2. Materials and Methods

### 2.1. Participants

Forty volunteers were recruited to take part in this experiment. All participants needed to be aged between 22 and 45 years old and have normal or corrected-to-normal vision. Experienced drivers needed to have had a driving license for at least 4 years; novices were required not to have a driving license. Twenty participants were assigned to the FCWS (FCWS) condition, and the remaining twenty to the No Automation (NA) condition.

The FCWS condition group was composed of (i) 10 experienced drivers (6 females) aged 23.3 years (±2.1) with driving experience of 5.1 years (±2.0) and an average declared annual driven kilometers of 9098 (±7476) and (ii) 10 novice, unlicensed drivers (four females) aged of 25.2 years (±4.3) without any driving experience. Two FCWS group participants (one experienced and one novice) dropped the experiment and were excluded from the analyses.

The NA condition group was composed of (i) 10 experienced drivers (6 females) aged 23.8 years (±3.4), having had a driving license for at least four years with driving experience of 5.6 years (±2.7) and an average declared annual driven kilometers of 9575 km (±6898) and (ii) 10 novice drivers (3 females), who were unlicensed drivers aged of 25.4 years (±3.3) without any driving experience.

The methods were carried out per the relevant guidelines and regulations and reviewed by an ethics committee; see the institutional review board statement below.

### 2.2. Equipment

#### 2.2.1. Driving Simulation

The experiment was run on a fixed-base simulator equipped with an automatic gearbox, providing a horizontal field of view of about 145° using three computer screens, with an analog speedometer available at the bottom-center of the middle screen. Apart from the screens, the experimental setup was composed of an adjustable seat (JCL Sim Racing), a steering wheel with force feedback, an accelerator, and brake pedals (Logitech G27). The driving simulation software was developed by the University of Sherbrooke; see [31] for more details.

#### 2.2.2. FCWS

For the FCWS group, a perfectly accurate FCWS was active all the time, and participants were not offered the possibility to switch it off. When the driven vehicle approached the leading vehicle too closely (time to collision below 3.5 s), an auditory warning was played (Figure 1A describes FCWS functional architecture). That FCW trigger threshold was selected based on previous research [32]. The warning sound was created following a criterion that showed superior efficiency compared with other FCW auditory designs [18]. As such, the warning sound lasted 3.5 s and contained 3 bursts of 4 fast beeps with inter-burst intervals of 250 ms. Its fundamental frequency was 319 Hz, and its duty cycle was a 5 s waveform; for more detail, see [18]. The auditory waveform was a D,F-M alert.

For the No Automation (NA) group, no warning was provided in case of a near-collision situation, everything else being equal.

#### 2.2.3. Eye-Tracking

An eye-tracker (iView X head-mounted; SensoMotoric Instruments) was used to record eye movements at a sampling rate of 50 Hz after a 9-point calibration procedure. The calibration accuracy was checked by experimenters to ensure that all calibration fixations made by participants were within a 2° arc around each of the 9 calibration points. Fixations and saccades were segmented using a velocity threshold algorithm I-VT [33] with a fixation velocity below 45°/s.

### 2.3. Procedure

After the presentation of the experiment, written informed consent was obtained from all participants. Participants of the FCWS group were informed that automation would be present, and its working principle was detailed.

Participants were then invited to drive either with automation (FCWS group) or without (No Automation group) for about six hours spread over six consecutive weeks. This allowed drivers to face a variety of simulated driving situations, such as driving on highways, navigating in town, and steering along bendy roads.

The FCWS was only relevant during the specific car-following task (a leading vehicle in front of the driven vehicle and, thus, possible near–forward collisions). At some point in the driving simulation, participants caught up with the leading vehicle and were prompted, through the central screen, to follow that vehicle at a close but safe distance, as if they were following friends and were unaware of the itinerary. The car-following task was run six times (i.e., once every hour of simulated driving or so) for a period of 5 min each time, representing a total of 30 min of car-following tasks per participant. The leading vehicle (i.e., vehicle to follow) speed was programmed to vary along a sine wave pattern oscillating between 50 kph and 80 kph with a period of 30 s. No oncoming traffic was present during the car-following task.

Drivers’ behaviors were exclusively analyzed during that car-following task.

### 2.4. Data Analysis

#### 2.4.1. Driving Behaviors

First, the total number of near-collision events was counted (time to collision with the leading vehicle < 3.5 s).

Second, for participants who did experience at least one near-collision event, behaviors during these events were investigated with three complementary metrics (see Figure 1B):-*Brake Reaction Time (BRT)*: the time between the beginning of a near-collision event (TTC < 3.5 s, warned in FCWS or unwarned in NA) and the first contact with the brake pedal.-*Amplitude of Brake Pedal Pressure (Amp BPP):* how much pressure was applied on the brake pedal, normalized between 0 and 1.-*Minimum Time To Collision (Min TTC)*: the minimal time to contact with the leading vehicle during the near-collision event.

Third, car-following performances outside the near-collision events were analyzed based on three criteria according to previous research [34,35]:-*Coherence:* a squared cross-correlation between the main frequencies of the leading vehicle and driving vehicle speeds (from 0 = no coherence to 1 = perfect coherence). The higher the value, the more the participants’ speed was adapted to the leading-vehicle speed changes.-*Delay(s)*: required time for participants to react to the leading-vehicle speed changes. The faster the reaction, the shorter the delay.-*Gain*: Amplitude of the driven vehicle speed changes compared to the leading-vehicle speed changes. A gain of 1 would indicate a driven vehicle speed change exactly proportional to leading vehicle speed change; a gain greater than 1 indicated an overreaction, whereas a gain smaller than 1 represented an underreaction).

#### 2.4.2. Visual Behaviors

Visual behaviors were investigated only during the near-collision events. Three complementary indicators were metrics:

First, the spread of search in both the horizontal (X) and vertical (Y) axes was computed using the standard deviation of fixations coordinates [36]. This indicator has been observed to be sensitive to the driving experience in non-automated driving [37,38].

Second, the mean number of fixations made by drivers to assess visual explorations of different locations.

Third, the mean fixation duration was measured, an indicator of the time taken to process gazed elements.

#### 2.4.3. Statistics

For each outcome, an ANOVA with two between-participant factors (automation (FCWS condition, NA condition) and driving experience (novice drivers, experienced drivers)) was performed. The level of significance was set at *p* < 0.05; trends are reported when *p* < 0.10.

## 3. Results

### 3.1. Driving Behaviors

#### 3.1.1. Number of Near-Collision Events

The ANOVA only revealed a significant main effect of FCWS (*F*(1,34) = 4.53, *p* < 0.05, *η*^2^ = 0.12; see Figure 2) on the number of near-collision events. On average, drivers faced more near-collisions with the leading vehicle in the control condition without automation (M = 7.05) than with FCWS (M = 2.5). This corresponds to 64.5% fewer near-collision events with an FCWS compared to a control condition without. In other words, the number of near-collision events without automation is divided by 2.82 when the FCWS is available.

No significant main effect of driving experience or interaction effects between experience and automation were found (*F*(1,34) = 0.51, *NS*, and *F*(1,34) = 0.01, *NS*, respectively).

#### 3.1.2. Behaviors during Near-Collision Events

##### Brake Reaction Time (BRT)

Mean BRTs were significantly different depending on the automation condition (*F*(1,26) = 8.31, *p* < 0.01, *η*^2^ = 0.24; see Figure 3). Longer BRTs were recorded in the FCWS than in the NA condition.

No significant main effect of driving experience or in interaction with automation was found (*F*(1,26) = 0.20, *NS*, and *F*(1,26) = 0.42, *NS*, respectively).

##### Amplitude of Brake Pedal Pressure (Amp BPP)

The ANOVA revealed a significant main effect of driving experience (*F*(1,26) = 7.60, *p* < 0.05, *η*^2^ = 0.20; see Figure 4) on the amplitude of brake pedal pressure (Amp BPP). A non-significant trend was also found for the automation condition (*F*(1,26) = 3.42, *p* < 0.05, *η*^2^ = 0.09; see Figure 4). Novice drivers pressed the brake pedal harder than experienced drivers, and the FCWS generated a trend toward harder pushes on the brake pedal than in the NA condition.

No significant interaction between automation conditions and the driving experience was found (*F*(1,26) = 0.48, *NS*).

##### Minimum Time to Collision (Min TTC)

A significant interaction between automation conditions and the driving experience was recorded (*F*(1,26) = 4.42, *p* < 0.05, *η*^2^ = 0.13; see Figure 5). Post-hoc analyses indicated a non-significant trend for shorter Min TTC for novice drivers in the NA condition compared to experienced drivers in the NA condition and to novice drivers in the FCWS condition (*p* < 0.1).

No significant main effect of automation condition or driving experience was found (*F*(1,26) = 1.74, *NS*, and *F*(1,26) = 1.19, *NS*, respectively).

#### 3.1.3. Car-Following Performances

The ANOVA revealed a significant effect of automation condition in terms of coherence (*F*(1,26) = 5.84, *p* < 0.05, *η*^2^ = 0.18; see Figure 6). The coherence was poorer in the FCWS condition compared to the NA condition. But, no significant effect of the automation condition was found for the delay and gain indicators (*F*(1,26) = 0.58, *NS*, and *F*(1,26) = 0.13, *NS*, respectively).

No significant effect of driving experience was observed for all three indicators (coherence, delay, and gain, respectively: *F*(1,26) = 0.27, *NS*; *F*(1,26) = 0.97, *NS*; and *F*(1,26) = 0.01, *NS*).

No significant interaction between automation condition and the driving experience was found for coherence and delay (*F*(1,26) = 824, *NS,* and *F*(1,26) = 1.01, *NS*, respectively), but a non-significant trend was observed for gain (*F*(1,26) = 3.51, *p* < 0.08, *η*^2^ = 0.12). Post-hoc analyses did not show any significant differences; furthermore, gain values were very close to 1 in all conditions (NA-E: 0.94; NA-N: 1.03; FCWS-E: 1.05; and FCWS-N: 0.95).

### 3.2. Gaze Behaviors

#### 3.2.1. Spread of Search

The ANOVAs revealed a significant effect of automation condition on the Y-axis (*F*(1,26) = 7.04, *p* < 0.05, *η*^2^ = 0.20) but not on the X-axis (*F*(1,26) = 2.54, *NS*). Along the Y-axis, a narrower spread of search was recorded in the FCWS condition (M = 1.06) than in the NA condition (M = 2.18).

No significant effect of driving experience (X-axis: *F*(1,26) = 0.20, *NS*; Y-axis: *F*(1,26) = 0.01, *NS*) or an interaction between the automation condition and driving experience (X-axis: *F*(1,26) = 0.64, *NS*; Y-axis: *F*(1,26) = 2.16, *NS*) were found.

#### 3.2.2. Mean Number of Fixations

The mean number of fixations was significantly different depending on the automation condition (*F*(1,26) = 5.93, *p* < 0.05, *η*^2^ = 0.18). Fewer fixations were made in the FCWS condition (M = 2.20) than in the NA condition (M = 3.17).

No significant effect of driving experience or interaction between driving experience and the automation condition was recorded (*F*(1,26) = 0.45, *NS*, and *F*(1,26) = 0.28, *NS*, respectively).

#### 3.2.3. Mean Fixation Duration

No significant effects of automation condition, driving experience, or the interaction between driving experience and automation condition were observed (respectively, *F*(1,26) = 2.24, *NS*; *F*(1,26) = 1.43, *NS*; and *F*(1,26) = 0.93, *NS*).

## 4. Discussion

The reported experiment aimed at investigating the FCWS on near–forward collision events for novice and experienced drivers. Also, visual and driving behaviors were collected for natural, unprovoked near-collision events in a car-following task using driving simulation. As compared to the control condition, participants in the FCWS condition were involved in about one-third of the number of near-collision events, corresponding to a drop of 64.5% in these risky events. Results showed a significant safety improvement in line with several previous observations made both under simulated and real-life experiments, e.g., [10,15,39]. Novice drivers did not take greater advantage of the FCWS presence than experienced drivers, as both groups of participants benefitted from near-collision event reduction in a similar fashion.

### 4.1. Driving and Visual Behaviors during Near–Forward Collision Events

Focusing on the near–forward collision events, both driving and visual measures were computed to understand drivers’ behaviors and associated cognitive processes. Unexpectedly, BRTs were observed to be significantly longer for warned near-collisions than for unwarned near-collisions, and despite a non-significant trend for a harder push on the brake pedal for warned near-collisions, no major reduction of the Min TTC was recorded for warned near-collisions. Altogether, these results indicate that the FCWS did not improve drivers’ responses during the near-collision events experienced in the simulated drives. When considering gaze behaviors, the FCWS did reduce the spread of search in the Y-axis dimension and the number of fixations compared to the NA condition. Gaze data are consistent with earlier observations [23,24,25,26] and could be interpreted as a visual-attentional focus on the relevant elements of the driving situation. Still, these visual strategies changed, and the associated improved visual perception of the situation did not transfer to more efficient driving behaviors. This might be due to the nature of the near-collision events faced by participants in the current study. The near-collision events were not provoked, for instance, using a secondary task [40], experimenting on drowsy drivers [29], or by inducting distraction [41]. Furthermore, the near-collision events were not as critical to safety as if the leading vehicle had braked strongly unexpectedly. Drivers might have approached the leading vehicle too closely, but the corresponding brake responses to be applied were not extremely urgent. Within this context, the FCWs could have been processed with a cognitive cost that translated into longer BRTs with the FCW than without. This extra cognitive cost did not impact the Min TTC thanks to a slightly stronger brake response observed through a non-significant trend of the brake pedal amplitude.

From a theoretical perspective, the findings suggest that FCWs did not simply trigger faster brake responses in the case of a warned near-collision. Rather, drivers engaged in tactical processes by analyzing the driving context [9,42]. Here, possibly because the near-collision events were not as safety-critical as in the case of a strong brake of the leading vehicle, drivers were not required to respond faster. In contrast, the warning process possibly came at a cognitive cost that slowed down the BRTs. This is of particular interest, as it challenges the rationale that warnings are simple, straightforward signals directly interpreted by drivers that can only facilitate the process of information. Here, the opposite was observed, supporting the idea that cognitive processing associated with a warning may be adaptable depending on the driving context [9].

### 4.2. Driving Experience Influence during Near–Forward Collision Events

Concerning the impact of driving experience on results, no significant differences were observed between novice and experienced drivers for the number of near-collisions, car-following behaviors, and gaze behaviors. The only influence of driving experience was recorded for driving behaviors during near-collision events, with novice drivers pressing the brake pedal harder and presenting shorter Min TTCs in the NA condition. More precisely, in the NA condition, novice drivers’ BRTs were not longer than those of experienced drivers, but they did show significantly more ample pressure on the brake pedal and still presented a trend of significantly shorter Min TTCs compared to those of experienced drivers. This indicates that, despite reacting as fast as experienced drivers and pressing the brake pedal harder, the avoidance maneuver was still less safe. A plausible explanation is that novice drivers did not regulate the braking response as smoothly as experienced drivers, possibly by pressing the brake pedal too gently or too harshly at the beginning of the braking phase and thus adjusting the pressure of the brake pedal unevenly. In contrast, experienced drivers regulated braking more smoothly. Note that this observation and related interpretation is only valid in the NA condition. In FCWS presence, the Min TTC values were not significantly different for novice and experienced drivers. This can be interpreted as an FCWS being an extra benefit for novice drivers. This extra safety benefit is not due to an improvement in gaze explorations and associated attentional redirection, as no significant differences between novice and experienced drivers were found in any of the gaze outcomes. It can, therefore, be explained by FCWS assistance to analyze the driving situation (i.e., an action must be performed) and possibly the nature of the action to be implemented (i.e., brake).

### 4.3. Driving Behaviors outside Near–Forward Collision Events

As shown in other studies [43], a limited, general behavioral adaptation has been observed. The presence of the FCWS has changed the driving behaviors outside the near–forward collision events warned. In other words, this effect can be understood as an indirect response of drivers to the presence of the FCWS. The behavioral adaptation was spotted in the car-following behaviors. During the experimental task, the quality of the car-following behaviors was investigated with three complementary metrics: the coherence, delay, and phase between the leading and the driven vehicle. Whatever the driving experience group, participants exhibited high coherence values in both the FCWS (M = 0.70) and the NA (M = 0.83) conditions. These results indicate that the car-following task was cautiously performed in both conditions. Regarding the coherence between the leading vehicle and the participants’ vehicle, a significant degradation was observed in the FCWS condition as compared to the NA control condition (with almost a 16% coherence drop). This indicates that the presence of the FCWS, even inactive, resulted in speed changes by the participants less adapted to the leading-vehicle speed changes. An interpretation is that under the FCWS condition, drivers would take an extra safety margin to avoid the trigger of FCWs. However, behavioral adaptation due to the FCWS presence appeared restricted to the coherence metric, as no significant differences were found for the delay, while gain values were very close to 1, showing that participants’ speed changes were almost exactly proportional to the leading-vehicle speed changes. Our findings are in line with the appearance of behavioral adaptations to FCWSs, as previously observed with LDWSs [44,45,46] and FCWSs [21,27]. It is difficult to judge firmly whether the behavioral adaptation is positive or negative, as a massive reduction of near-collision events was recorded (positive effect), whereas the quality of the car-following task decreased (negative effect).

## 5. Conclusions

While FCWSs have been commercially available now for several years and tend to be more and more common in real cars, there is still work to undertake to explore in detail how drivers interact with such automation devices. Future investigations are welcome to investigate the influence of the driving context on FCWS efficiency. The emergency of a near-collision situation can be manipulated through the speed of the TTC reduction with the leading vehicle, for instance. In the current study, we selected participants who were unexperimented, as the novice drivers included in the study did not hold a driving license. Still, this level of expertise is based on initial training and does not consider actual individual performance. Another perspective can be to classify driving expertise based on actual driving performances rather than initial training and driving experience. Interestingly, warning automation has a long-lasting tradition in the car-driving domain, e.g., [47], and such warning automation, apparently simple and well understood, still requires researchers’ attention to better describe its impact on drivers’ behaviors and safety benefits. Ironically, even the current most advanced automated vehicles simply warn drivers when resuming manual control is required [48].

## Figures and Tables

**Figure 1 sensors-23-08150-f001:**
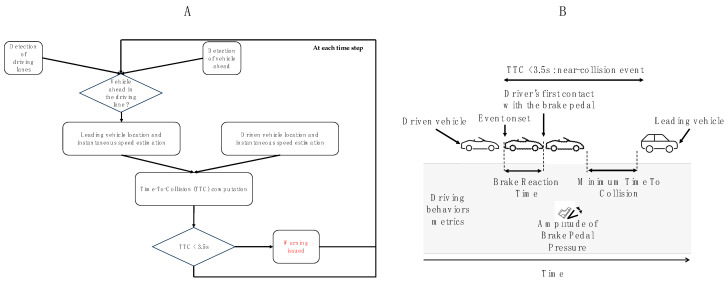
(**A**). Functional architecture of the FCWS. (**B**). Schematic description of a near-collision event over time, along with the driving variables metrics computed (dotted vehicles represent the driven vehicle location over time).

**Figure 2 sensors-23-08150-f002:**
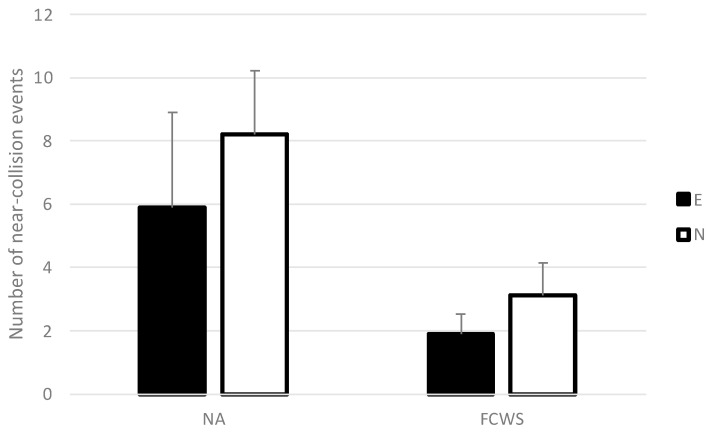
Total number of near-collision events in NA and FCWS conditions for experienced and novice drivers. Error bars represent one standard error.

**Figure 3 sensors-23-08150-f003:**
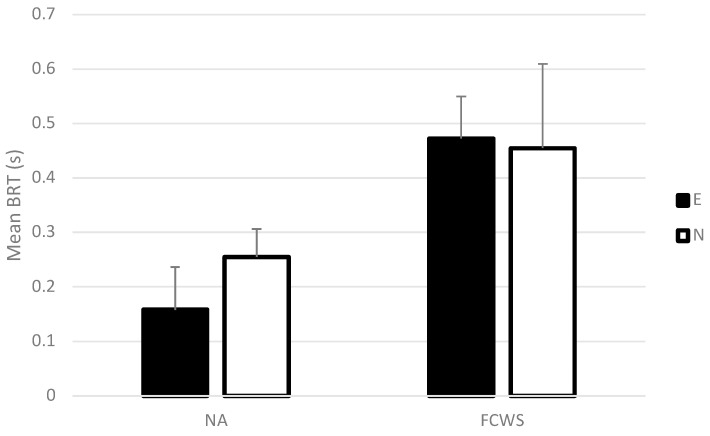
Mean brake reaction time in NA and FCWS conditions for experienced and novice drivers. Error bars represent one standard error.

**Figure 4 sensors-23-08150-f004:**
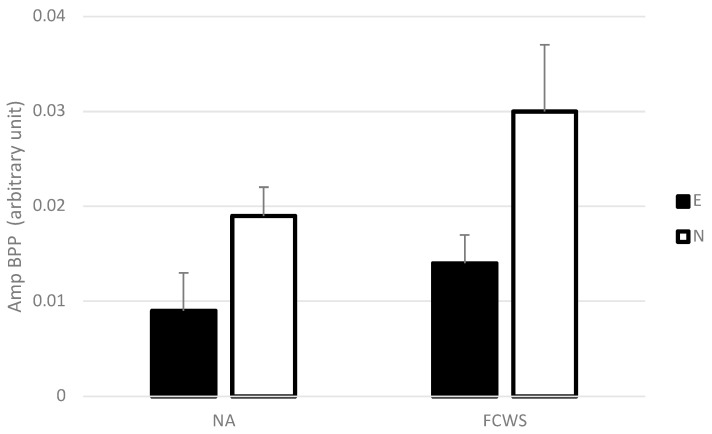
Mean Amplitude of brake pedal pressure in NA and FCWS conditions for experienced and novice drivers. Error bars represent one standard error.

**Figure 5 sensors-23-08150-f005:**
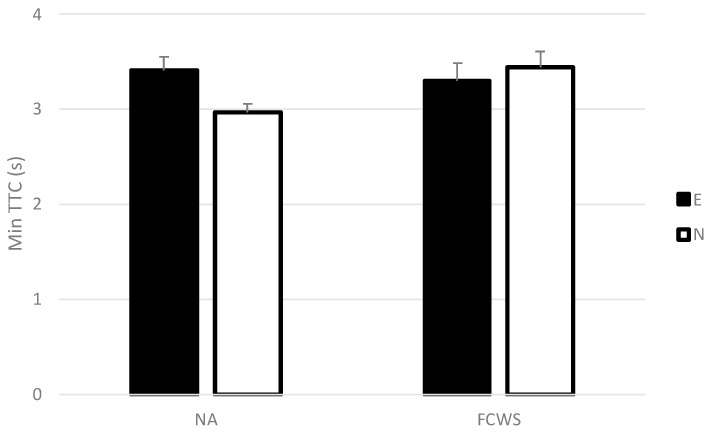
Mean minimum time to collision in NA and FCWS conditions for experienced and novice drivers. Error bars represent one standard error.

**Figure 6 sensors-23-08150-f006:**
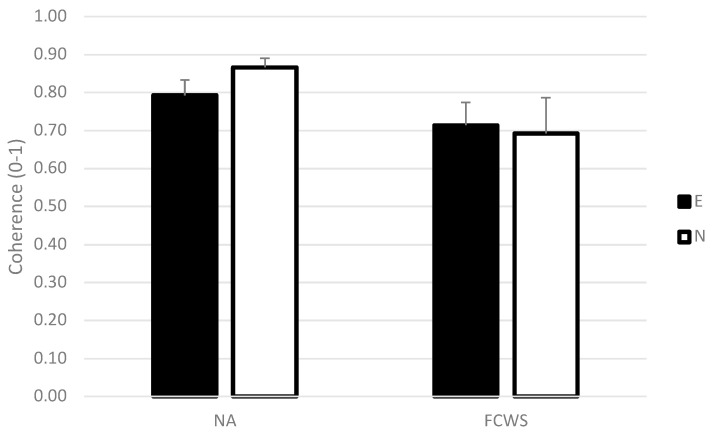
Mean coherence between the lead vehicle and the driven vehicle in NA and FCWS conditions for experienced and novice drivers. Error bars represent one standard error.

## Data Availability

Data are available from the corresponding author upon request.

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
