# Peer review of "Comparison of Experienced and Novice Drivers’ Visual and Driving Behaviors during Warned or Unwarned Near–Forward Collisions"

_sensors, 2023, doi:10.3390/s23198150_

Round 1

Reviewer 1 Report

1. It is suggested that the abstract should be rewritten to highlight the experimental characteristics of this study and to summarize the conclusions more succinctly.

2. We suggest to describe the FCWS functional architecture (line 159) and the FCWS test scenario (line 198) with pictures to enrich the content of the article.

3. The fourth section is lack of analysis theory (line 326) . The subjectivity of this part is strong, but lack of theoretical support. The collision time TTC algorithm is determined by the relative distance between the vehicle and the obstacle in front (usually the vehicle) and the average relative velocity. From my point of view, combined with the experimental theory of experimental data for further analysis can make the article more convincing.

4. The fourth section is not organized systematically (line326) , which can be described by sections in the order of the third chapter.

Author Response

R1.1. We followed R1’s suggestion, the abstract has been rewritten to (i) better highlight the experimental characteristics and (ii) provide a clearer summary of the conclusions.

R1.2. In line with R1’s comment, a figure (see Figure 1) has been added to the manuscript to better describe the functional architecture of the FCWS and a near-collision event.

R1. 3. We are not sure to fully understand R1’s remark. TTC or Time To Collision is indeed the time it would take for a collision to occur at an instantaneous speed, and distance of both the driver’s and the leading vehicle. This is a standard metrics since its original definition by Hayward (1972), which stated that TTC was: "The time required for two vehicles to collide if they continue at their present speed and on the same path". It is widely used in research now, see for example Borsos et al. (2020), Kiefer et al (2006) or Wessels et al (2022). We are aware that details of calculation of TTC can evolve since the ones proposed by Van der Horst & Hogema (1994) (see for example Hou et al (2014)) but our computation remained simple if not basic, since assessing TTC validity was not the main point of our contribution.

  • Borsos, A., Farah, H., Laureshyn, A. & Hagenzieker,M. (2020) Are collision and crossing course surrogate safety indicators transferable? A probability based approach using extreme value theory, Accident Analysis & Prevention, 143, https://doi.org/10.1016/j.aap.2020.105517.
  • Hayward, J.Ch. (1972). Near miss determination through use of a scale of danger. Report no.TTSC 7115, The Pennsylvania State University, Pennsylvania
  • Hou, J., List, G. F., & Guo, X. (2014). New Algorithms for Computing the Time-to-Collision in Freeway Traffic Simulation Models. Computational Intelligence and Neuroscience, 2014, 761047. https://doi.org/10.1155/2014/761047
  • Kiefer, R. J., Flannagan, C. A., & Jerome, C. J. (2006). Time-to-Collision Judgments Under Realistic Driving Conditions. Human Factors, 48(2), 334–345. https://doi.org/10.1518/001872006777724499
  • van der horst, Richard & Hogema, Jeroen. (1994). Time-To-Collision and collision avoidance systems. Proceedings of 6th ICTCT workshop, Salzburg
  • Wessels, M., Zähme, C. & Oberfeld, D. Auditory Information Improves Time-to-collision Estimation for Accelerating Vehicles. Curr Psychol (2022). https://doi.org/10.1007/s12144-022-03375-6

R1. 4. We would like to thank R1 for the cautious reading of the submitted paper and the insights provided. We are not certain to get this point, but we would like to state that, as it is common use in the cognitive ergonomics domains and more generally in cognitive sciences, we organized the discussion section (referred as the ‘fourth section” ) as following in our original submission :

Paragraph 1: Synthesis of the experimental manipulations and main results collected

Paragraphs 2&3: Driving and visual behaviours during near-forward events

Paragraph 4: Influence of driving experience

Paragraph 5: Driving behaviours outside near-forward collision events

Paragraph 6 : Limitations, perspectives and conclusions

In this part of the paper, authors are usually expected to discuss the reported results based on previously available research and provide interpretations of the findings. Therefore, the point of view of the authors is important here and the interpretation proposed should not be restricted to the results themselves. This organisation allows us to discuss the results along the different independent variables (i.e., experimental manipulations) according to the different measures recorded during the experiment. This indeed requires the reader to follow a different structure that the one use for the previous “results” section, but we considered that, nevertheless, this structure was suited for  discussing the different aspects of the reported research rather than the dependant variables (i.e., behavioural, and visual metrics) used to investigate the topic of interest. Therefore Results and Discussion do not follow the same structure, as it seemed to us that this would diminish the potential interest of the Discussion section.

As our organization was possibly confusing, we decided, in line with R1’s comment and improve the readability of the submitted manuscript, to add subtitles in different sections of the discussion. We sincerely hope that these subtitles will allow the reader to capture the structure of the discussion at a glance and makes this structure transparent.

Reviewer 2 Report

Authors have investigated visual and driving behavious of drivers during near-forward collisions. The drivers are divided to experinced and novice groups. Results show that the Forward Collision Warning System greatly reduced the number of near-collision events at a cost of longer reaction time. The experiment is well designed and results are clearly presented and discussed. I would like to recommend for publication.

Author Response

R3. We would like to warmly thank Reviewer 3 for their very positive appreciation on the submitted manuscript.